# Advancements in Wearable and Implantable Intraocular Pressure Biosensors for Ophthalmology: A Comprehensive Review

**DOI:** 10.3390/mi14101915

**Published:** 2023-10-09

**Authors:** Kevin Y. Wu, Mina Mina, Marjorie Carbonneau, Michael Marchand, Simon D. Tran

**Affiliations:** 1Department of Surgery, Division of Ophthalmology, University of Sherbrooke, Sherbrooke, QC J1G 2E8, Canada; yang.wu@usherbrooke.ca (K.Y.W.);; 2Department of Mechanical and Manufacturing Engineering, University of Calgary, Calgary, AB T2N 1N4, Canada; 3Faculty of Dental Medicine and Oral Health Sciences, McGill University, Montreal, QC H3A 1G1, Canada

**Keywords:** miniaturized biosensors, biomedical research, nanomedicine, wearable electronics and sensors, glaucoma, intraocular pressure (IOP), implantable sensors, engineering approaches, clinical integration, ophthalmology

## Abstract

Glaucoma, marked by its intricate association with intraocular pressure (IOP), stands as a predominant cause of non-reversible vision loss. In this review, the physiological relevance of IOP is detailed, alongside its potential pathological consequences. The review further delves into innovative engineering solutions for IOP monitoring, highlighting the latest advancements in wearable and implantable sensors and their potential in enhancing glaucoma management. These technological innovations are interwoven with clinical practice, underscoring their real-world applications, patient-centered strategies, and the prospects for future development in IOP control. By synthesizing theoretical concepts, technological innovations, and practical clinical insights, this review contributes a cohesive and comprehensive perspective on the IOP biosensor’s role in glaucoma, serving as a reference for ophthalmological researchers, clinicians, and professionals.

## 1. Introduction

Glaucoma’s insidious nature and its complex interplay with intraocular pressure (IOP) present a multifaceted challenge that continues to captivate and perplex the ophthalmologic community. This disease, also known as the “silent thief of sight”, stands as one of the leading causes of irreversdible blindness worldwide. While the role of IOP in glaucoma has long been recognized, recent advancements and novel perspectives have emerged, necessitating a comprehensive re-evaluation.

An emerging frontier in this re-evaluation is the development and integration of IOP biosensors. These cutting-edge technological tools enable precise and continuous monitoring of IOP, opening new avenues for understanding, diagnosing, and managing glaucoma. The advent of wearable and implantable IOP sensors signifies a potential pivotal shift in glaucoma care, combining technological innovation with patient-centered solutions.

This review article presents a pioneering examination of intraocular pressure (IOP) sensors, reflecting a remarkable convergence of insights across pathophysiology, engineering, and clinical practice. Beginning with an in-depth exploration of IOP’s significance, including both its essential role in eye health and its function as a risk factor for glaucoma, this review emphasizes the potential benefits of continuous IOP monitoring. This recognition paves the way for a comprehensive analysis of cutting-edge IOP sensors from an engineering perspective. The review delves into the latest technological innovations in wearable and implantable IOP sensors, offering a detailed overview of their design, functionality, and potential in complementing glaucoma care. These biosensors marry technological innovation with patient-centered solutions. This review further integrates these sensor technologies into the clinical landscape, elaborating on real-world applications, patient-centered strategies, and future directions. The synthesis of these domains contributes to a comprehensive and innovative perspective on IOP sensors, underscoring the novelty of this review.

By synthesizing research from 2017 to 2023 and offering insights from pathophysiology to the engineering aspect, and then to the practical clinical applications of IOP sensors, this review not only contributes to the current body of knowledge but also paves the way for future innovations and the potential for complementing existing glaucoma management.

## 2. Overview of Glaucoma

### 2.1. Understanding the Importance of Intraocular Pressure and Its Role in Glaucoma

Positive intraocular pressure (IOP) is vital for the eye’s functionality and health. Firstly, the eye is a semi-rigid sphere, and positive IOP helps maintain its shape and structural integrity. Secondly, the eye requires a constant flow of nutrients for its tissues, and IOP assists in regulating this nutrient supply, thereby supporting the eye’s metabolic needs.

However, while positive IOP is crucial for eye health, excessively high IOP becomes detrimental, putting undue pressure on the optic nerve head. Over time, this excessive pressure can cause damage to the optic nerve head, leading to the development of glaucoma [1].

Glaucoma, ominously termed the “silent thief of sight”, stealthily progresses without initial notable symptoms, often leading to delayed diagnosis and intervention. This condition, as the principal cause of irreversible blindness globally, is characterized by glaucomatous optic neuropathy, distinctively marked by the cupping or excavation of the optic disc, axonal degeneration, apoptosis of retinal ganglion cells, and subsequent irreversible vision loss [1]. The disease’s multifactorial etiology encompasses both genetic and environmental elements. IOP is a continuous risk factor for glaucoma development, although it is not invariably elevated in primary open-angle glaucoma (POAG) patients, thus not defining the disease. Ocular hypertension (OHT) signifies elevated IOP without glaucomatous visual field or optic disc damage, and notably, not all patients with OHT eventually develop glaucoma [2].

IOP typically exhibits a Gaussian distribution in young adults (20–40 years old), with an average value of 15.5 d 2.6 mmHg [3]. However, this distribution becomes skewed in older populations, exhibiting a non-Gaussian tail at higher IOP levels (>40 years old). The 97.5th percentile for IOP falls at 22 mmHg, and an IOP within 2 standard deviations of the mean (i.e., 12 to 21 mmHg) is generally considered “normal” [4,5]. Traditionally, glaucoma was defined as an IOP greater than 21 mmHg, essentially a statistical abnormality within the general population [2,3,4]. This definition has proven erroneous as glaucomatous damage can also occur within the “normal” IOP range. Therefore, clinicians should avoid imprecisely labeling elevated IOP conditions without confirmed glaucomatous neuropathy as ‘glaucoma’.

As such, it is essential to dispel the common misconception that elevated IOP serves as the defining characteristic of glaucoma. Instead, IOP should be recognized more accurately as one of many risk factors contributing to glaucoma’s multifaceted etiology [6,7]. Other risk factors include, but are not limited to, the excavation of the optic disc, peripapillary hemorrhage, thin corneal thickness, low corneal hysteresis, age, family history, and certain syndromes such as pigmentary dispersion and pseudoexfoliation [8]. Other systemic conditions that could lead to decreased ocular perfusion pressure, such as migraines, nocturnal hypotension, and sleep apnea, have also been implicated in the disease’s onset and evolution [8].

Despite this multitude of risk factors, it is noteworthy that IOP is currently the only modifiable risk factor within the direct sphere of influence for ophthalmologists. Therefore, managing IOP remains the cornerstone of glaucoma treatment strategies, emphasizing its importance. Building on this foundation, this article will further delve into the contemporary approaches of managing IOP, specifically focusing on the advancements in continuous IOP monitoring facilitated by innovative IOP sensors.

### 2.2. Global Prevalence of Glaucoma

Glaucoma, as the principal cause of irreversible blindness globally, represents a significant public health challenge. As of 2020, it is estimated that glaucoma affects approximately 80 million individuals worldwide, with about 11.2 million suffering bilateral blindness [9]. The global prevalence sits at roughly 3.5% among individuals aged 40–80 [9,10]. The increasing prevalence is driven by an aging population and longer life expectancies, suggesting a sharp rise in glaucoma cases in the upcoming decades [9,10].

### 2.3. Classifications of Adult Glaucoma

Adult glaucoma is classified as open angle or angle closure, and as primary or secondary. Open-angle glaucoma (OAG), marked by no visible trabecular meshwork obstruction, is differentiated into primary (POAG) with no clear IOP elevation cause, and secondary with an identifiable etiology. Normal-tension glaucoma is a disputed term for POAG without apparent IOP elevation.

Angle-closure glaucoma, featuring trabecular meshwork obstruction by the peripheral iris, is subdivided into primary and secondary, based on the presence of underlying disease [11].

Primary open-angle glaucoma (POAG) and primary angle-closure glaucoma (PACG) are associated with distinct and context-dependent risk factors. POAG’s critical risk elements include elevated intraocular pressure (IOP), diminished ocular perfusion pressure, advanced age, thinner cornea, high myopia, and race, ethnicity, and genetic susceptibilities [12]. Conversely, PACG prevalence is influenced by age, race, ethnicity, and a potential gender predisposition, with a higher prevalence often observed in women. Hyperopia and a family history of angle closure are also pivotal [13,14].

### 2.4. The Dynamics of Aqueous Humor: Production, Outflow, and Their Role in Intraocular Pressure

The aqueous humor, a fluid essential for eye function, is produced in the posterior chamber by the ciliary processes at a rate of 2–3 μL/min when awake, roughly halving during sleep [15]. Key to this process are the nonpigmented epithelial cells rich in mitochondria and microvilli. Aqueous humor production involves active secretion, ultrafiltration, and simple diffusion, with the former being the most significant process and involving the enzyme carbonic anhydrase II [15].

Various drugs can suppress the formation of aqueous humor, potentially through the inhibition of carbonic anhydrase or the blockade of β2-receptors, impacting active secretion [15].

Aqueous humor decreases in production as we age. Interestingly, despite this decrease, the incidence of primary open-angle glaucoma (POAG) rises with age. This suggests that the increased IOP in glaucoma may not be due to overproduction of aqueous humor, but is mainly secondary to impaired outflow of aqueous humor [15,16].

The outflow of aqueous humor involves two major mechanisms: the pressure-sensitive trabecular pathway and the pressure-insensitive uveoscleral pathway [15,16]. The former is facilitated through the trabecular meshwork and Schlemm’s canal, eventually returning the fluid to the systemic circulation [16]. The uveoscleral pathway is more ambiguous but involves passage through the ciliary muscle bundles and the supraciliary and suprachoroidal spaces [16].

The modified Goldmann equation (Equation (1)) provides a simplified model for understanding intraocular pressure (IOP) dynamics [17]:(1)$$P_{0}=\left(\frac{F-U}{C}\right)+P_{v}$$
where $P_{0}$ is the IOP; F is the rate of aqueous humor production; U is the uveoscleral outflow rate; C is the trabecular outflow facility; and $P_{v}$ represents the episcleral venous pressure. Essentially, IOP is determined by the balance between aqueous humor production and its outflow, with the outflow being a sum of the pressure-insensitive uveoscleral pathway (U) and the pressure-sensitive trabecular pathway (C), adjusted for the episcleral venous pressure ($P_{v}$). In simplified terms, it captures how pressure within the eye is a function of fluid production, outflow, and external venous pressure. Outflow resistance is the inverse of the outflow facility, with a higher resistance implying a lower facility, leading to elevated IOP [17].

## 3. Measurement of IOP

Tonometry, the non-invasive method for measuring intraocular pressure (IOP), encompasses various techniques, each with its unique strengths and limitations. Despite their utility, all current methods carry inherent inaccuracies and none can reliably measure the pressure inside the eye chamber in all instances [18].

Applanation tonometry is the most widespread method for measuring IOP. This technique works according to the Imbert–Fick principle (Equation (2)), which assesses IOP by measuring the force required to flatten a specific area of the cornea [17,18,19,20].
(2)$$P=\frac{F}{A}$$

The pressure (P) inside a thin-walled sphere equals the force (F) required to flatten its surface divided by the area (A) of the flattening.

The Goldmann applanation tonometer, a widely recognized and commonly used instrument in this category, serves as the gold standard for measuring IOP. It operates by gauging the force necessary to flatten a specified area of the cornea, precisely 3.06 mm in diameter [21]. This specific diameter is chosen as the resistance from the cornea to flattening is counterbalanced by the capillary pull from the tear film meniscus. This device also uses a split-image prism allowing for accurate measurement of the flattened area [21]. To delineate the area of flattening, fluorescein dye and topical anesthetic are added to the tear film.

However, several factors can influence the accuracy of this method. For instance, an insufficient amount of fluorescein can lead to poor visualization, thereby affecting the measurement. Tear film thickness can also impact the reading; a thick tear film can result in a falsely high IOP reading due to lower surface tension, while a very thin film may yield a falsely low IOP reading due to higher surface tension [22].

Astigmatism can produce inaccurate readings as the fluorescein pattern appears elliptical, causing IOP readings to be artificially high or low [17]. Other corneal characteristics such as edema, scarring, and thickness can also influence the reading. Specifically, central corneal thickness (CCT) significantly impacts IOP measurements. Readings are most accurate when the CCT is 520 μm [17,23]. Thicker corneas resist deformation leading to overestimated IOP, while thinner corneas may give artificially low readings [17,18].

However, the relationship between CCT and measured IOP is not linear, and the biomechanical properties of individual corneas, such as stiffness or elasticity, can further affect IOP measurement [17,23]. There is currently no validated correction factor for the effect of CCT on applanation tonometers. In addition, it is worth noting that a thin central cornea is a recognized risk factor for progression from ocular hypertension to glaucoma [17,18,23,24].

Various other tonometers employ distinct methodologies to measure IOP. Each technique carries its own set of strengths and limitations, which affects their applicability. This typically hinges on the specific condition of the individual patient, as well as the resources available for the procedure. For more detailed information, please refer to Table 1.
Mackay–Marg–type tonometers: these flatten a small area of the cornea and measure the pressure in the center of the ring; devices like Tono-Pen are portable and useful for patients with corneal scars or edema, as well as those who can only be in a supine position, despite possible inaccuracies at high and low IOP levels (i.e., overestimate low IOPs and underestimate high IOPs) [17,25,26].Pneumatonometer: shares some characteristics with Mackay–Marg–type devices. It applanates the cornea until a gap closes in a constant flow of air, making it useful for eyes with corneal scars, edema, or keratoprostheses, and is able to monitor IOP continuously [17,27,28].Noncontact tonometers (air-puff): measure the force of air required to flatten the cornea, often overestimating IOP but are valuable for large-scale glaucoma-screening programs; the Ocular Response Analyzer (ORA) improves accuracy by using correction algorithms [17,29,30,31].Rebound tonometers: measure the speed of a small probe’s deceleration and rebound after impacting the cornea, suitable for pediatrics and home tonometry, but significantly influenced by central corneal thickness [17,32,33].Dynamic contour tonometer: a nonapplanation contact tonometer measures IOP by aligning the cornea’s surface with the instrument tip, believed to be less influenced by corneal properties and thickness (Schneider & Grehn, 2006) [17,34].Indentation tonometry (Schiøtz): measures corneal indentation produced by a known weight, accuracy is highly dependent on ocular biomechanical properties, and does not require electrical power [17].

Tactile tension: involves digital pressure on the globe to estimate IOP. It may be inaccurate but is useful in uncooperative patients, in cases involving keratoprostheses, or for detecting large differences in IOP between eyes [17].

## 4. Intraocular Pressure (IOP) Fluctuation

Intraocular pressure (IOP) fluctuation, though recognized, remains debatable as an independent risk factor for glaucoma. IOP fluctuations are categorized into ‘instantaneous’ (seconds, due to saccades, blinking); ‘diurnal–nocturnal’ (daily, influenced by physiological and environmental changes); and ‘short-term’ and ‘long-term’ (spanning weeks to years) [35]. Evaluating the influence of these variations on glaucoma progression is complex, with several studies yielding diverse conclusions. This section seeks to explore the interplay between IOP fluctuations and glaucoma progression, underscoring the potential significance of continuous IOP monitoring using sensors, and offering a more nuanced understanding of IOP fluctuations relevant to glaucoma progression [35].

### 4.1. Instantaneous IOP Fluctuation

Although transient IOP spikes occur during eye movements, blinks, and eye rubbing, direct evidence linking them to glaucoma progression remains elusive. Animal studies suggest these abrupt elevations may affect the optic nerve. However, their definitive contribution to glaucoma, particularly in high-strain eyes, requires further investigation [36,37,38,39].

### 4.2. Diurnal-Nocturnal IOP Fluctuation

Evidence indicates diurnal–nocturnal IOP fluctuations potentially influence glaucoma progression [40,41,42,43].

In a recent study by De Moraes et al., the authors investigated the relationship between 24 h IOP changes and the rate of glaucoma progression [44]. By utilizing a contact lens sensor to monitor IOP for 24 h in forty patients, they discovered that larger IOP peaks (over 90 mV) and the average speed and magnitude of these peaks were strong predictors of faster glaucoma progression. Importantly, a single day of monitoring with this sensor yielded more informative results about glaucoma progression than multiple IOP measurements taken during office visits [44].

Although IOP usually peaks at night, these increases might not harm the optic nerve. The rise in IOP at night is mainly due to the supine sleeping position, which can enhance eye perfusion and potentially counterbalance the increased IOP. Similarly, an increase in cerebrospinal fluid pressure, common in this position, could offset IOP-induced stress by reducing the pressure difference across the optic nerve. The body may also have additional mechanisms to cope with regular biorhythms associated with IOP and blood pressure [45].

Taking these considerations into account, it is plausible to suggest that daily fluctuations in IOP, in conjunction with systemic blood pressure and ocular blood flow variations, may have a significant role in the onset and progression of glaucoma. Therefore, the interplay between fluctuating IOP and a compromised regulation of ocular blood flow could potentially endanger retinal ganglion cells that are susceptible to changes in ocular perfusion pressure, leading to axoplasmic stasis and subsequently apoptosis [45]. In light of these theories, even though further evidence is required for validation, the potential importance of continuous IOP monitoring with sensors is increasingly apparent. This continuous monitoring could be useful both in a research setting, for a better understanding of the disease dynamics, and in clinical practice, allowing for personalized treatment regimens based on each patient’s unique IOP fluctuation patterns [45].

### 4.3. Short-Term IOP Fluctuation

Short-term fluctuations in IOP, occurring over days to weeks, do not directly impact glaucoma progression but could predict future long-term fluctuations. In an investigation by Japanese researchers [46], they measured short-term (24 h) IOP changes using a Triggerfish^®^ contact lens sensor (CLS), and long-term fluctuations over a mean period of 5 years from routine clinical records. Parameters such as mean, difference, standard deviation, and peak of IOP were assessed. The results revealed a significant correlation between short-term CLS readings and long-term clinical data, suggesting that 24 h IOP monitoring could aid in forecasting long-term IOP changes. This finding suggests a potential benefit of short-term IOP monitoring in anticipating long-term IOP fluctuations; however, further research is required to validate its implications in glaucoma management [46].

### 4.4. Long-Term IOP Fluctuation

Multiple studies have established a connection between long-term IOP fluctuation, obtained from inter-visit measurements, and the progression of glaucoma [47,48,49,50,51,52,53,54,55]. Notably, the Advanced Glaucoma Intervention Study (AGIS) and the Collaborative Initial Glaucoma Treatment Study (CIGTS) indicated that IOP fluctuation, as opposed to the mean IOP, was correlated with the progression of glaucoma, particularly in instances with low mean IOPs [49,56]. The Japanese Archive of Multicentral Databases in Glaucoma (JAMDIG) study found similar associations [51].

However, a contrasting perspective emerged from studies such as the Ocular Hypertension Treatment Study (OHTS), Early Manifest Glaucoma Treatment (EMGT), European Glaucoma Prevention Study (EGPS), and Ocular Hypertension Treatment in Diagnostic Innovations in Glaucoma Study (DIGS). These found no significant influence of IOP fluctuation on visual field (VF) progression [57,58,59,60].

It is postulated that long-term IOP fluctuation could cause damage by disrupting physiological rhythmic cycles and inducing irregular stress on the optic nerve head. Therefore, managing IOP fluctuations, particularly in patients with low mean IOPs, might be helpful. This emphasizes the potential utility of continuous IOP monitoring via sensors, offering a more comprehensive understanding of IOP fluctuations and their potential impacts on glaucoma management.

### 4.5. Beyond IOP Reduction: The Importance of Stability in Glaucoma Management

In light of these considerations, the clinical management of glaucoma patients may need a shift in focus from merely reducing intraocular pressure (IOP) to modulating it [35]. This quality-based approach towards IOP control, aiming for stability over time, could potentially be more effective in preventing disease progression than simply lowering the IOP quantity. In primary open-angle glaucoma (POAG) cases that continue to progress, maintaining a steady IOP, while curbing high peaks, could be as important as achieving low IOP levels, especially for patients whose disease progresses at low mean IOP levels [35]. As a result, the growing recognition of the need for stable IOP modulation underscores the potential significance of continuous IOP monitoring using sensors. These devices could offer valuable insights into IOP fluctuations, providing a more comprehensive view of glaucoma’s progression in both research and clinical contexts. This observation, even though requiring further validation, highlights a promising direction in glaucoma management.

## 5. Principles and Engineering Aspects of IOP Biosensors

### 5.1. Engineering of Contact-Lens-Based Sensors

Contact-lens-based sensors (CLBS) for monitoring intraocular pressure (IOP) can be categorized into three main types based on their sensing principles, namely, optical, microfluidic, and electrical sensors. Regardless of their categorization, however, one central concept is fundamental to the operation of all CLBS [61]. This concept is the relationship between the IOP and the corneal curvature.

In a recent study, Campigotto et al. used 12 cadaveric eyes and varied the IOP from 10 to 36 mmHg, as expected in the case of glaucoma [61]. The authors observed that the deformation from the curvature of the eye—as it varies within the specified range—was significant enough to be captured and mimicked by a silicone contact lens overlying the eye. This deformation of the contact lens is the foundational block for all CLBS. For instance, a rudimentary design of a CLBS is demonstrated in Figure 1. As can be seen, with the application of a stressor, in the form of increased IOP, there is a change in the dimensions and shape of holes, which can then be observed and correlated to that IOP change. In the figure, it can be observed that the peripheral circular holes are transformed into ellipses when the IOP increases [62]. While such a simple design is theoretically effective, this specific design may lack the sensitivity needed to measure relatively small fluctuations in IOP, such as those seen in glaucoma.

Although the study by Campigotto et al. demonstrates the capability of CLBS to detect changes in IOP, critics of the technology may argue that variations in anatomy and physiology could affect the general applicability of these findings. For instance, corneal stiffness can be impacted by biochemical properties including central corneal thickness (CCT), radius of corneal curvature (RCC), corneal hysteresis, hydration, and age [63,64]. Critics can then argue that increasing corneal stiffness would decrease the deformation in the curvature of the eye, and thus render CLBS without effect. However, in another study, researchers studied physiological variations in CCT and RCC and determined that these differences did not impact the measurements made with contact lens sensor [64]. The two variables, CCT and RCC, were specifically selected since these two variables were considered to be the most significant factors that can impact IOP measurement [64].

#### 5.1.1. Microfluidic Dilatometer-Based Wearable Sensors

While many microfluidic sensors have been designed for monitoring IOP [65,66,67,68,69,70], these sensors function based on one main mechanism—converting strain to measurable fluid displacement. To understand how this mechanism works, a free body diagram displaying the forces acting on a contact lens is shown in Figure 2. It can be seen that five major forces act on the contact lens [71]. *P*_1_, which is the hydrostatic pressure, acts by transmitting the IOP through the tear film that is between the cornea and the contact lens. Exerting a force in the opposite direction to *P*_1_ is the atmospheric pressure, *P*_0_. The surface tension cohesive forces acting between the tear film and the lens are represented as *F_σ_*. Together, *P*_0_ and *F_σ_* hold the contact lens in place, attaching it to the cornea. *F_lid_* is the force acting on the lens when blinking, and W is the weight of the contact lens exerted due to the force of gravity. With these forces at equilibrium, contact lenses can comply with corneal deformations, acting like a hollow hemisphere shell. Consequently, when IOP increases and the corneal curvature changes, only a circumferential tensile strain is exerted on the contact lens [72]. Converting this circumferential tensile strain to a fluid displacement is the foundational working mechanism of microfluidic sensors, as shown in Figure 3.

As seen in Figure 3, the contact lens is often designed with a sensing reservoir and a display microchannel. With the increase in IOP, the stretching of the contact lens increases the sensing reservoir volume, having a suction effect and drawing the air–liquid interface toward the sensing reservoir. The reverse effect is seen with a decrease in IOP—the contact lens is released from tension decreasing the sensing reservoir volume and thereby pushing the air–liquid interface further away from the sensing reservoir. This displacement of the air–liquid interface is detectable, and is often designed to be monitored through cellphone imaging by the patient or their supports [65,68]. The displacement, $D_{\varepsilon }$, defined in relation to the strain, ε, is described by Equation (3):(3)$$D_{\varepsilon }=\frac{∆V_{\varepsilon }}{A}$$
where $∆V_{\varepsilon }$ is the change in volume; and A is the cross-sectional area of the sensing reservoir [65].

Microfluidic sensors are advantageous—they create the opportunity for continuous, noninvasive monitoring of IOP that is accessible widely, without the need for advanced imaging, expert interpretation, or expensive equipment. However, one major disadvantage of these systems is that they do not permit 24 h IOP monitoring without disruption of the sleep cycle since the user needs to be awake for imaging of the sensor. This disadvantage has encouraged innovation in the field to develop other solutions that are capable of autonomously monitoring and transmitting IOP data directly to smartphones.

To determine the potential of commercializing microfluidic CLBS, a patent search was conducted. The search revealed that it was only recently, in 2021, that a patent application was granted to Smartlens Inc. (Mountain View, CA, USA), showing the novelty in the field and perhaps indicating the shift to commercialization soon [73].

#### 5.1.2. Optical (Phototonic Crystal) Wearable Sensors

The naturally occurring structural color of chameleon skins [74], butterfly wings [75], and flying birds [76] has attracted the fascination of scientists for many decades. Structural color, often termed as physical color, is defined by the intricate relationship between incident light and the submicro- or nanostructures of the material with which it interacts [77]. It is dependent on fundamental optical principles such as those of refraction, scattering, and diffraction [78], and is independent of chemical pigments or organic dyes [79]. Structural color has been implemented for IOP monitoring through the manufacturing of CLBS with colloidal photonic crystals (PCs) [80,81,82,83]. The spatial periodicity of PCs provides them with their chromatic properties; the periodic arrangement prevents the propagation of certain wavelengths of visible light, thereby, reflecting structural color perceptible by the naked eye [84]. This range of photon frequencies or wavelengths of electromagnetic waves that are prevented from propagating are labeled as the “photonic forbidden band” or “photonic band gaps” (PBG). It is the change in the PC structure and the resulting impact on the PBG that allows chameleons to shift color based on their skin tension [85].

The complex interaction between the light and the lattice structure of PCs follows Bragg’s law [Equation (4)]:(4)$$m\lambda =2nd\cdot \sin\left(\theta \right)$$
where m is the order of diffraction; λ is the wavelength; n is the average refractive index of the colloid and the surrounding matrix; d is the lattice spacing; and θ is the angle between the incident light and diffracting crystal planes. From Equation (4), it can be observed that a decrease in the lattice spacing, d, would result in a shorter wavelength. With increased IOP and stretching of the contact lens, the lattice spacing decreases, and the CLBS would demonstrate a color change towards shorter wavelengths, such as the blue color (Figure 4).

PCs can be coated on the surface of hydrogels [86], the water-containing polymers that are currently used in the design of FDA-approved soft contact lenses [87]. Hydrogels react to small changes in pressure, resulting in a deformation that is translated to the superimposed PC structure, thereby leading to photonic effects [88]. Over the recent years, research with photonic crystals has prepared them to meet market demands. As can be seen from Equation (4), photonic crystals generate an iridescent effect—where the angle of illumination causes distinct hue changes in the contact lens. While this optical phenomenon is typically beautiful to behold, as with soap bubbles reflecting a spectrum of light, it can create ambiguity when it is used for IOP monitoring. However, with innovative material developments, scientists have been able find symmetrical microspheres, such as SiO_2_, which can reflect the same chromaticity in a large viewing angle [84].

One disadvantage with PC-based sensors is the limited wavelength shift in the reflectance spectra. That is, the deformations are typically small resulting in minor wavelength and color changes that require high-precision instruments to measure them. To overcome this weakness, researchers have come up with innovative solutions. For instance, Maeng et al. used a microhydraulic system to amplify the deflection of the PC layer, making use of Pascal’s principle (Equation (5)).
(5)$$\delta_{s}=\delta_{c}\cdot \left(\frac{A_{c}}{A_{s}}\right)$$
where $\delta_{s}$ is the deformation of the PC membrane; $\delta_{c}$ is the deformation of the contact surface; and $\frac{A_{c}}{A_{s}}$ is the ratio of surface areas. From this equation, it is evident that the deformation in the PC membrane is amplified due to the large $\frac{A_{c}}{A_{s}}$ ratio.

Another solution to improve the sensitivity of optical CLBS has been the design of dual-sensing smart contact lenses that can additionally monitor key tear fluid biomarkers. For instance, Ye and colleagues developed a structural color contact lens that detects changes in IOP, that can additionally detect matrix metalloproteinase-9 (MMP-9) [82]. MMP-9 is a tear biomarker that shows increased activity in early glaucoma [89]. Consequently, it can be used as a predictive biomarker for glaucoma, alongside IOP monitoring, thus, creating dual sensing for the target disease.

#### 5.1.3. Electrical Wearable Sensor

Electrical CLBS can be divided based on two main working mechanisms, namely, piezoresistive and inductive couple telemetry sensors [80]. These sensors are discussed in the subsequent sections, highlighting the advantages and disadvantages of each.

##### Piezoresistive Sensors

Piezoresistive sensors function by converting a mechanical stimulus, such as changes in corneal curvature secondary to IOP fluctuation, to changes in electrical resistance. The piezoresistive sensing mechanism is based on the properties of the sensor material. When strain is applied, the electrically conductive network increases in resistance due to disconnections or minor breaks in the fabric of the sensor material, as seen in Figure 5. Many contact lenses have recently been developed that function based on the piezoresistive properties [90,91,92,93,94,95]. In contrast to optical-based contact lenses, the electrical detection approach experiences fewer disruptions due to eyelid movements, and is thus considered to be more reliable [93]. Central to these sensors is an understanding of Wheatstone bridge—an electrical circuit that is used for determining the value of an unidentified electrical resistance. In the case of the contact lens, this unknown electrical resistance is first calibrated to an IOP measurement establishing a relationship between electrical resistance and IOP. This, then, allows for monitoring of the IOP by following the electrical resistance relative to its initial state. Governing this sensing principle is Equation (6):(6)$$R=\frac{\rho l}{A}$$
where the resistance of the material, R, is dependent on the resistivity (ρ), the length (l), and the cross-sectional area (A). As seen through the equation, small strains resulting from changes in IOP cause high-density cracks in the sensing material, which then increases the material’s resistivity and length [95]. Similarly, when the material recovers, it results in a decreased length and thus, decreased resistance [95]. Figure 5 demonstrates this working principle.

The primary engineering consideration of piezoresistive types of sensors is based on the material selection and the associated fabrication process. A variety of materials have been proposed, including silver nanowires [90,94], graphene nanowalls [93], reduced graphene oxide and carbon nanotubes [91], and graphene woven fabrics [95]. The diversity in materials chosen for this application over the past five years reflects that no single material is yet considered ideal. However, as can be seen, graphene is often selected to be part of the material due to its favorable fracture behavior, transparency, and biocompatibility [95].

##### Inductive Couple Telemetry Sensors

In inductive couple telemetry sensors, a variety of sensing units are used to convert mechanical inputs to resonant frequency shifts [80]. These include inductive, capacitive, and resistive sensing units [96,97,98]. Once the mechanical input is converted, the frequency shift is then read outside by an inductive coupling link [80]. These sensing units have been a major focus of attention in the field for the past decade. Their mechanisms are well understood and described abundantly in the literature. In the recent years, developments have focused on improving specific outcomes. One such outcome is designing soft and comfortable contact lenses; thus, researchers have introduced new materials, such as room temperature liquid Galinstan to attain such an outcome [99]. Another desirable outcome has been to improve the sensitivity of the sensors and reduce the measurement error gag, where again, researchers have made substantial leaps forwards [100]. Such improvements over the years will surely drive commercialization and improvement in current approved IOP sensors, such as the ones described in later sections.

### 5.2. Engineering of Glasses-Based Sensors

Recently, in addition to the developments in contact lens-based sensors, scientists and engineers have aspired to create non-contact and non-invasive gadgets that can be applied for IOP monitoring. One such innovation has been wearable glasses that can deduce IOP by observing the radius of curvature of a grid pattern applied to the cornea [101]. The glasses use the concept of optical triangulation to monitor IOP. The whole device consists of a laser, multiple mirrors and lenses, a mask—which creates the desired grid on the cornea—and a miniaturized camera that is used to monitor the grid [101]. By following the grid pattern, it is possible to determine fluctuations in IOP [101].

Similar to these glasses-based models, other designs have been proposed that incorporate glasses with an implantable element. One example of this is a diffraction grating interferometric implantable micro-electromechanical systems (MEMS) sensor, that is incorporated with readout glasses (The MEMS Sensor-Glasses Pair for Real-Time Monitoring of Intraocular Pressure by Zolfaghari et al. [102]). The combination of glasses with implantable sensors reflects the innovation that the field is undergoing, and indicates the progress being made.

### 5.3. Engineering of Implantable Sensor

As compared to wearable biosensors, implantable IOP sensors typically require surgical intervention for insertion. The following sections focus on a few of these sensors with a focus on the novelty achieved over the past few years.

#### 5.3.1. Capacitor-Based Sensors

Like CLBS, scientists have also developed capacitor-based sensors that are implantable. One such device is designed for implantation in the superotemporal quadrant of the eye between the sclera and conjunctiva [103]. Generally, these devices consist of a few main components. The first component is a commercial pressure sensor with a suitable range, that can detect the IOP. Assembled with the pressure sensor is a chip that supports wireless power/data telemetry. Once assembled, these components are submerged in a biocompatible material to minimize the risk of adverse inflammation reactions. Finally, the chip uses a radiofrequency coil to receive power and transmit measurement to an external reader [103]. These components can be seen in Figure 6.

#### 5.3.2. Implantable Fabry–Perot Pressure Sensor for Keratoprosthesis

Keratoprostheses (KPro) surgery is an effective milestone in treating corneal blindness [104]. However, while improvements in the surgical technique have minimized most short-term complications, unfortunately, glaucoma continues to persist as a significant and potentially severe issue for individuals who have undergone keratoplasty [105]. Adding to the complexity of IOP management for post keratoplasty patients is that it is not feasible to monitor IOP by GAT or other rebound tonometry due to the rigid B-KPro implant [106]. Consequently, the IOP is often estimated by finger palpation [106].

To improve IOP in such cases, researchers have developed a fiber-optic pressure sensor, as seen in Figure 7. The working mechanism of this sensor is based on a Fabry–Perot optomechanical cavity [107]. This cavity is formed between two parallel surfaces with a separation length denoted as ‘L’ [107]. The first surface is a smooth glass surface, while the second surface, responsible for sensing, is constructed from a thin and flexible silicon membrane. This membrane undergoes deformation in response to changes in pressure within the anterior chamber of the eye, changing the cavity length from L to L’. Due to this change in cavity length, interference results in interferometric fringes in the reflected light, which can then be correlated to IOP through an optical analyzer [107].

#### 5.3.3. Microfluidic-Implantable Sensors

Implantable microfluidic devices use the same principle as wearable microfluidic sensors. As seen in Figure 8, the IOP sensor consists of an internal gas reservoir that is connected to the intraocular liquid through a microfluidic channel. Through capillary forces and intraocular pressure, the microchannel is filled with liquid, thereby compressing the gas within the reservoir until both gas and liquid pressures achieve equilibrium [108]. Heightening the IOP leads the interface to move towards the gas reservoir, while reducing the IOP causes the interface to shift towards the channel’s entrance [108]. While the ideal placement of this IOP sensor would be to integrate it with an intraocular lens commonly used in cataract surgery, it would also be possible to independently implant the device [108]. One of the advantages of this device is that the pressure can be read through a smartphone camera, like some of the CLBS. However, in this case, due to the internal placement of the device, an adapter and an image analysis software would be needed to accurately determine the IOP [108].

While unique, this sensor has not gained much attention in literature or in patents. Possibly, this lack of traction can be attributed to the success of its CLBS counterparts. Even with regards to the sensitivity, CLBS showed sensitivities of 660 μm/mmHg [70] and 600 μm/mmHg [109], which were much higher than the implantable counterpart, which only achieved a sensitivity of 137 μm/mm Hg [108].

## 6. Comparison of the Different Sensors

Though an initial, superficial comparison between wearable and implantable devices might incline one towards the appeal of contact lens sensors because of their non-invasive advantages, a more thorough examination unveils a different perspective. As can be seen in Figure 9, in general, implantable devices tend to have higher sensitivity than comparable contact-lens-based counterparts [110].

## 7. Clinical Trials, Commercialization, and Advancements in the Field

### 7.1. Contact Lens-Based Sensors

#### 7.1.1. Sensimed Triggerfish (Triggerfish CLS, Sensimed AG, Lausanne, Switzerland)

One of the prominent advances in the field of IOP monitoring was the FDA approval of the Sensimed Triggerfish contact lens sensor in 2016 (Triggerfish CLS, Sensimed AG, Lausanne, Switzerland) [111,112,113]. The Triggerfish CLBS is a silicone contact lens that detects changes in the corneoscleral limbus curvature due to changes in IOP variations [114]. The sensor records at 5 min intervals over the 24 h period, giving the user 288 readings a day [115]. The Triggerfish has been extensively studied both in clinical trials and otherwise [116,116]. The CLBS has been tested for possible influences of electromagnetic radiation on its measurement behavior [117] and has been used to predict the visual field loss [44,118,119]. Overall, it has been concluded by a recent narrative review of all studies involving Triggerfish that the device is a promising tool that, while undeniably beneficial, still has to be further refined and improved [114].

The FDA approval and commercialization of Triggerfish CLBS has broadened the prospects for research, improved the patient experience, and deepened ophthalmologists’ comprehension of IOP. For instance, the Triggerfish was used to monitor the impact of daily activities such as exercise, yoga, and emotional stress [120]. The results showed a statistically significant IOP-related profile elevation due to resistance training and emotional stress [120]. Such results create the opportunity to enhance patient outcomes by regulating the medication prescription or by promoting activities that can decrease IOP. Another benefit of the Triggerfish is its potential application in teleophthalmology [121]. Over the first few months of the COVID-19 pandemic, there was a reduction of 81% in medical visits including an approximately 88% reduction in glaucoma visits [122]. While there may be no way to offer the full range of testing yet, providing IOP measurements may be one of the first steps in enabling glaucoma care virtually [122]. ‘Teleglaucoma’ care would be an invaluable asset for those in rural areas, those with barriers to transportation, and those who may otherwise be confined due to illness.

All in all, the Triggerfish presents numerous advantages. However, it can also lead to a few adverse effects, especially over a long period of wearing time—such as causing transient myopia [114]. With further iterations, these adverse events can be reduced, increasing the benefit–risk ratio of the device.

#### 7.1.2. Glakolens Inc.

Similar to the Triggerrfish, Glakolens (Istanbul, Turkey) is developing a novel, non-invasive CLBS. Their commercialization efforts, evident through their website and their US Patent (US10067075B2), demonstrate the working principle of their product. It is evident from these that their device is composed of a metallic resonator that is placed on the contact lens. This resonator undergoes a frequency shift under mechanical stress like that exerted by an increase in IOP. The frequency shift is then translated by the device to reflect changes in IOP. Consequently, the sensor transmits the detected changes wirelessly to a Holter-monitor-like device which records the data for the duration of the 24 h. The product, which is worn by the patient for a day, is then returned to a clinic for evaluation by an ophthalmologist.

### 7.2. Implantable IOP Sensors

#### 7.2.1. EYEMATE, Implandata

Even more recently, the EYEMATE-SC IOP pressure sensor (Implandata Ophthalmic Products GmbH, Hannover, Germany) has been trialed in clinical trials (NCT03756662). These most recent trials are of a device that has been iteratively improved and tested—both in animal models, as well as human trials [123,124,125]—over the past decade.

The first-generation device was tested in the ARGOS-01 study, where six patients with open-angle glaucoma and cataract were enrolled, implanted with the device, and followed for a year [123]. The initial results showed promise; the IOP measurements showed similar profiles compared to those of Goldmann applanation tonometry [123]. Additionally, these trials showed, for the first time, that it was possible to continuously monitor IOP in glaucoma patients. The results of the ten year follow-up of the ARGOS-01 study were recently published, and the authors concluded, with almost 25,000 IOP measurements, good long-term safety, tolerability, and functionality of the implanted sensors [126]. However, while implanting the wireless telemetric sensors into the sulcus of the eye was found to be mostly safe and capable of offering consistent, long-term IOP measurements, this approach presented with some drawbacks. The main issues relate to irritation and thinning of the iris, pupillary distortion, postoperative anterior chamber inflammation, and pigment dispersion [127]. These factors collectively prompted a redesign of the sensor’s size and shape [127].

Similar to its predecessor, the second-generation device, branded as the EYEMATE-IO, was implanted within the ciliary sulcus (ARGOS-02 trial) [106,124,128,129]. The device measures IOP by an array of capacitive pressure sensors, which detect capacitance change and transmit it externally by radiofrequency [130]. An external handheld reader allows for IOP measurements to be taken as needed, while also providing wireless power to the device [124]. The new design was both smaller in diameter and thinner, aiming to reduce the devices’ footprint and safety concerns [127]. The implantation and testing, which was performed on 22 participants, showed that all device-related adverse events resolved quickly with appropriate treatment [124]. Moreover, comparative measurements with GAT showed good agreement [124]. According to these published findings, the EYEMATE-IO system was granted regulatory clearance within the European Union (CE mark) for its application in patients diagnosed with open-angle glaucoma [124]. Beyond showing the efficacy of the device, and granting regulatory clearance, the results from the study confirmed theories that were previously postulated. For instance, in an analysis of the data obtained from the ARGOS-02 trial, daily and seasonal variations of IOP were observed. IOP was significantly lower on Fridays as compared to other weekdays, and was significantly higher in winter as compared with summer months [131]. Such a finding holds important clinical value and reflects on the benefits of having a continuous IOP monitoring device. Specifically, these findings indicate the potential to individualize titration of glaucoma medications to ensure that each patient has a customized regimen to manage their IOP fluctuation, over days, weeks, and seasons.

With the many advantages of the EYEMATE-IO device, one specific hindrance was noted—an indication for cataract surgery is a prerequisite for implanting the sulcus-placed microsensor [132]. Due to this prerequisite, the implantation often excludes younger age groups, where IOP monitoring can be of essential value. Additionally, concerns arise regarding potential obstruction of the iridocorneal angle and the subsequent exacerbation of intraocular pressure (IOP) due to the device’s placement [132]. In response to these concerns, the EYEMATE-SC device was developed which is implanted in the suprachoroidal space instead. By conducting animal studies [132], followed by human trials [125,133], the results are favorable, showing effectiveness of the device and a good safety profile at 6- and 12-month periods. Additionally, this device can be implanted in younger individuals who may not have an indication for cataracts. While these results are hopeful, in the future, larger and more heterogenous studies should be performed to confirm the results of Implandata’s devices and their generalizability.

An examination of Implandata’s devices provides insight into the developmental progress made over the past decade and affirms the devices’ potential, encompassing both technological advancements and regulatory approval prospects.

#### 7.2.2. IOP Connect (Injectsense, Inc.; Emeryville, CA, USA)

A unique device that has recently received the Breakthrough Device Program (BDP) designation from the FDA is Injectsense’s continuous IOP monitoring system, IOP Connect™ [134]. This IOP sensor is smaller than a grain of rice and is to be implanted via injection [121]. Patients are directed to wear specialized smart eyewear on a weekly basis to recharge the sensor and retrieve the recorded IOP data. The data are subsequently transmitted to a cloud-based database accessible to physicians. As of now, this device is exclusively designated for investigational purposes, but the patents issued by Injectsense Inc., and the BDP designation are hopeful indicators for the initiation of human trials in the near future [135].

## 8. Clinical Integration of Advanced IOP Biosensors

With respect to the clinical applicability of IOP biosensors, numerous papers—some as early as the work conducted in 2011 by Liu et al. [136]—have explored practical considerations of this emerging technology. This section comprehensively synthesizes the rationale, indications, workflow, implications, and challenges with which clinicians need concern themselves in the application of advanced IOP biosensors in the current healthcare landscape.

The onset of the 2019 Coronavirus pandemic marked a significant turning point for numerous emerging medical technologies. This period led to the commencement of what Ricur et al. subsequently referred to as the “COVID-19 Tsunami” [137]. During this period of social isolation and virtual-appointment adoption, the clinical utility of both wearable and implantable IOP biosensors was made manifest. One central example was the widespread acceptance and integration of “teleophthalmology” models that hybridize synchronous and asynchronous vision care modalities [137]. Real-time IOP monitoring via implantable and wearable biosensors was then the logical and practical extension of the clinical continuum that has seen ophthalmic assessment transition from strict clinician measurement, to hybrid virtual and synchronous examinations [137,138,139], and finally to a workable model of fully asynchronous IOP measurement. This change has received considerable attention in the literature in recent years and has the potential to transform clinical outcomes of IOP-dependent ophthalmic ailments such as glaucoma progression and visual field loss [140,141,142].

### 8.1. Advantages of Adopting Advanced IOP Biosensors

The argument for advanced IOP biosensors in clinical practice can be broken down into “demand side” factors, i.e., evolving clinical expectations and standards, and “supply side” factors, i.e., enhanced biosensor technology production and integration. These are discussed here in turn.

On the demand side, there is an increasing recognition among clinicians and researchers of the blind spots, so to speak, inherent to traditional office-based IOP measurement modalities. Specifically, discrete IOP measurements interspersed throughout the year [125] preclude the assessment of circadian and positional changes in IOP as well as intraday IOP maxima and minima [133,143] that would otherwise constitute a more representative clinical picture of a patient’s overall IOP. The clinical implications of random IOP measurement throughout the year are twofold. First, given that the size of diurnal IOP fluctuations is second only to IOP itself as a determinant of glaucoma progression, it is not practicable for traditional IOP measurement modalities to capture nocturnal IOP patterns and impossible to do so without waking the patient and introducing artefacts into the measurement [136,142,144]. Second, and as a corollary, Szurman et al. highlight that critical glaucoma treatment decisions are consequently made on the basis of limited and situational IOP measurements that are amenable to distortion by variables that are not accounted for during in-office measurements [125].

On the supply side, recent innovations in both CLS [114,145,146,147,148,149,150,151] and hydrogel contact lens technology [96,152,153] have enabled advanced biosensors to engage in accurate, safe, and clinically actionable 24 h monitoring of IOP in both normal and glaucomatous eyes. With respect to the fungibility of CLS data with GAT measurements, Szurman and colleagues [133] measured a mean difference in IOP of −0.2 mmHg between patients using an implantable CLS and those subject to GAT measurement, which the authors categorized as “excellent” and in excess of the instrumental agreement between non-biosensor-based IOP measurement modalities such as GAT as compared to rebound tonometry. With respect to inter-patient consistency, Mansouri and Gillman [154] reported greater inter-eye IOP measurement agreement among patients using a CLS as compared with standard tonometry. Accuracy data for IOP biosensors are summarized in Figure 10.

Critical to IOP measurement accuracy—and to the rationale for IOP biosensors itself—are advancements in hydrogel technology that have enabled the manufacture of soft, comfortable, and sensitive inductor–capacitor–resistor pressure sensors in the form of contact lenses [96,152,153]. Hydrogels are 3D polymers whose biocompatibility with human tissue make them optimal carriers for miniaturized instruments, such as IOP sensors [152]. From the first application of hydrogel-based biosensors as early as 1994, key advances in material science, wireless sensor technology, and inductive coupling technology have yielded the current generation of advanced biosensors that have reached commercial viability and provide highly sensitive continuous IOP measurement [81,152,155,156,157,158,159]. Looking forward, Qiao and colleagues [152] view hydrogel-based contact lens biosensors as integral pieces of medicine’s progression to A.I. and a machine learning-based approach to diagnostics. The triad innovations in soft electronics, signal detection, and algorithmic learning make it possible for IOP biosensors to capture data that can intelligently guide treatment decisions, predict treatment responses, involve patients in their own ocular health, and decrease progressive vision loss due to glaucoma, taking eye care closer to the theragnostic ideal to which it aspires.

### 8.2. Potential Indications of Advanced IOP Biosensors—The “When”

In addition to rendering practicable 24 h IOP measurements, i.e., when patients are asleep and in various postural positions in day-to-day life, there are specific clinical situations in which advanced IOP biosensors may constitute a venerable tool in the clinician’s armamentarium.

Given that elevated IOP is simultaneously the main driver of glaucoma and the central therapeutic target in clinical practice, wearable and implantable IOP biosensors are patently useful in glaucoma management [160]. The vast majority of studies using IOP biosensors are conducted upon patients with primary open-angle glaucoma (POAG) [124,143,148,151,161,162,163,164] and normal tension glaucoma (NTG) [143,148,151,162].

As an illustration of the clinical utility of wearable IOP biosensors, the ENIGMA study, in which Kim et al. [151] tracked the nyctohemeral, i.e., night (nycto) and day (hemera), IOPs in POAG and NTG patients was made feasible by a wearable and non-invasive IOP biosensor contact lens. Among the novel nyctohemeral trends observed, the authors found that peak IOP occurred during the nocturnal period in both POAG and NTG patients, whilst NTG patients manifested greater 24 h IOP variability overall [151]. Consequently, advanced biosensors offer an exciting new instrument by which to interrogate IOP in way that is vastly more comprehensive than GAT. Numerous studies demonstrate how CLS are enabling clinicians to (i) better identify individuals with normal in-office pressures but who actually experience IOP spikes nocturnally [119,136,143,144,150,151,164,165,166,167,168] and thus better risk-stratify and treat patients, (ii) document changes in IOP in response to both surgical [125,133,156,169] and pharmacological [114,143,150,170,171] glaucoma treatment, and (iii) triangulate IOP trends with other concurrent biometric data such as blood pressure and sleep position [151,172] for clinical research purposes.

IOP biosensors may also play a pivotal role in detecting acute changes in IOP, particularly in scenarios that require careful monitoring for drastic fluctuations. Trabeculectomy surgery, a glaucoma treatment that involves creating a drainage channel to release fluid from the eye, can result in significant post-operative IOP variation. This fluctuation may even reach 0 mmHg (hypotonia), potentially leading to choroidal detachment, requiring timely treatment. On the other hand, secondary glaucoma conditions, such as traumatic hyphema, uveitic glaucoma (Posner–Schlossman syndrome), or steroid response, can cause IOP to soar rapidly from normal levels to excessively high pressures (above 50 mmHg), constituting an ophthalmic emergency. In these instances, where clinicians might anticipate the possibility of severe IOP peaks in the ensuing days, IOP biosensors can be strategically employed to monitor the fluctuations. Even though there are no studies at this time investigating these indications, we can extrapolate that this utilization of IOP biosensors not only ensures prompt detection of these serious conditions but also aids in the timely initiation of the necessary therapeutic interventions, thereby enhancing patient care and preventing complications.

Building on these advancements, recent findings also underscore the broader implications of IOP biosensors in glaucoma research (availability of physical activity tracking data from wearable devices for patients with glaucoma). For example, advanced IOP biosensors have greatly improved our understanding of glaucoma and OSA. Muniesa et al. [165] found nocturnal spikes in IOP in OSA patients using a CLS, especially among those on CPAP therapy. This aligns with previous studies linking OSA severity to IOP elevation [166]. By exploring previously unknown aspects of IOP, these biosensors open new avenues in clinical research and practice.

### 8.3. Integrating IOP Biosensors into Clinical Workflow—The “How”

Beyond the exploratory applications of IOP biosensors discussed previously, there is considerable latitude in which to integrate CLSs into the current clinical landscape.

There are two main types of IOP biosensors amenable to practical clinical deployment. The first is the wearable IOP biosensor and the second is the implantable IOP biosensor. Wearable IOP biosensors, appearing in the literature as early as 2011 [168], are practically consistent with ophthalmic contact lenses and are have both empirical and practical patient [68,91,93,160,173,174,175,176,177] applications. The Triggerfish (AG, Switzerland) is currently the most widely used wearable IOP biosensor with sensor readings comparable to GAT in both accuracy and precision across both glaucomatous [120,145,146,150,151,154,161,162,167,168,172,178] and non-glaucomatous patients [151,172,178] and with excellent tolerability and minimal adverse effects [126,147]. Within the context of normal clinical workflow, wearable IOP biosensors are typically deployed in either confirmed or suspected glaucoma patients for the purposes of 24 h IOP monitoring, with the resulting data helping clinicians diagnose otherwise “borderline” or suspected cases of glaucoma [166], better risk-stratify confirmed glaucoma patients, and assess responses to glaucoma treatment [91,93,158,161,179,180].

Implantable IOP biosensors are generally placed when glaucoma surgery is indicated and can be performed either following surgery [156] or intra-operatively. The EYEMATE (Implandata Ophthalmic Products GmbH, Hannover, Germany) is the most common implantable IOP biosensor and exists either as an intraocular sensor, the EYEMATE-IO, or a suprachoroidally placed sensor, the EYEMATE-SC [124,125,133]. There are several other implantable IOP biosensors currently in various stages of development and/or testing [124,126,128,131,136,139,140,155,173,181,182,183,184]. Specifically, implantable IOP biosensors have been studied in the context of concurrent non-penetrating glaucoma surgery (NPGS) [125,133,169], minimally invasive glaucoma surgery (MIGS) [169], and selective laser trabeculoplasty (SLT) [146]. Interestingly, a number of studies have also situated implantable IOP biosensors within the context of cataract extraction [124,159,169,177] and posterior segment [184,185,186] surgeries, reflecting the multimodal applicability of IOP biosensors beyond glaucoma and portending an even broader usage case for both implantable and wearable IOP biosensors in the future.

An emerging application of IOP biosensors are as platforms for intraocular drug delivery [114,149,153,180,182,187,188,189]. Kim and colleagues [180] report a novel gold hollow-nanowire-based wearable IOP biosensor design with integrated drug delivery functionality. In this application, timolol, an IOP-lowering drug, is automatically administered intraocularly via a negative feedback system whereby IOP readings above a given setpoint trigger on-demand drug administration, in turn lowering IOP to safe levels and ensuring IOP spikes are reflexively dissipated and an optimal IOP is maintained [180]. This innovative integration of IOP monitoring with automatic drug delivery functionality not only overcomes the challenges of patient compliance by eliminating reliance on manual administration, but also establishes a finely tuned control system that prevents IOP peaks, ensuring a consistently stable and optimal intraocular pressure level. Such a system is particularly suitable for elderly individuals living alone, who cannot otherwise administer glaucoma medication by themselves or face difficulties in attending regular ophthalmologists’ follow-ups for IOP monitoring, thereby enhancing continuity of care. Herein crystallizes the theragnostic ideal of glaucoma management to which current IOP measurement and treatment modalities aspire and which advanced IOP biosensors are within striking distance.

### 8.4. Impact on Prognostic Outcomes and Patient Follow-Up—The “What”

The “downstream” utility of advanced IOP biosensors emerges in two ways: (i) in the prediction of disease and visual field progression and (ii) the integration of biosensor data with traditional in-office IOP measurement. Prognostically, Tojo and Hayashi [167] discuss the application of an IOP biosensor as a means of predicting the success of downstream trabeculectomy surgery. Specifically, the authors posit that post-operative IOP fluctuations during the nocturnal period, as measured continuously with a CLS, can predict trabeculectomy surgery outcome, with a lower absolute fluctuation portending improved prognoses [167].

With regards to glaucoma progression itself, De Moraes [44] demonstrated how data collected by the SENSIMED Triggerfish^®^, a CLS discussed above, can effectively predict visual field loss in glaucoma patients receiving treatment by measuring parameters including the number of IOP peaks and peak duration in a 24 h period. This continuous monitoring forms an ophthalmic ‘fingerprint’ for individual patients, enabling clinicians to tailor glaucoma treatment to decelerate visual field progression [119,144,190]. This annex represents an exciting application of advanced IOP biosensors as an instrument of precision medicine, allowing practitioners to move beyond treating merely an office-based IOP reading.

### 8.5. Challenges

As with any clinical innovation, there have been and remain challenges inherent to the clinical integration of IOP biosensors that warrant a discussion of the current state of clinical palatability of IOP biosensors. With respect to sensor-specific clinical complications, in their 2021 observational study of 24 participants using the SENSIMED Triggerfish^®^ CLS, Sharma and Ong [150] reported a compliance rate of 100% with no reported complications. Additionally, Mansouri and Gillman [154] reported greater inter-eye IOP measurement agreement among patients using a CLS as compared with GAT-acquired IOPs; critically, this study did report conjunctival hyperemia and blurred vision as mild complications of the biosensor that are consistent with findings by Otsuka and colleagues [191]. In their suprachoroidal application of an implantable IOP sensor, Szurman et al. [133] reported no serious device complications, such as device dislocation or migration, in all study participants. In the same study, Szurman et al. [133] examined the possibility of intraoperative complications associated with IOP biosensors, such as injury to the eye due to difficulties in manually handling the sensor or incorrect sensor orientation; however, they reported no instances of these complications. Other assessments of complications associated with both implantable and wearable IOP biosensors largely reflect the same trend of minimal to no serious adverse effects or indications for discontinuation of CLS with isolated cases of mild complications such as discomfort and blurred vision [124,126,128,133,150,191].

Beyond the acute complications associated with the act of placing IOP biosensors, several studies have examined the challenges associated with the wearing of long-term IOP biosensors. Toshida [192] reported a transient decrease in visual acuity and aggravation of astigmatism after 24 h continuous wearing of a SENSIMED Triggerfish^®^ CLS due to the orthokeratological profile of the contact lens, which resulted in a so-called “central island” corneal topography. Miki and colleagues [193] echo these findings and reported an increase in refractive error of approximately 18% after a 24 h CLS wear period. As a corrective, 3D measurements of patients’ corneal curvature have been proposed in order to better dimension the wearable biosensor to the specific geometric and refractive specifications of the user and thus reduce the reported problems associated with surface contact mismatch between the cornea and CLS [137,193].

## 9. Conclusions

This review has embarked on an exploration into the multifaceted world of intraocular pressure (IOP) sensors, illuminating their role in understanding and managing glaucoma. By synthesizing research from 2017 to 2023 and offering insights from pathophysiology to engineering, and then to the practical clinical applications of these sensors, we have uncovered a transformative pathway in glaucoma care. The advent of wearable and implantable IOP sensors represents a significant advancement, bridging technological innovation with real-world applications and patient-centered strategies. Yet, challenges still remain, underscoring the need for continued research and collaboration. This comprehensive examination not only enriches the current body of knowledge but also lays the foundation for future innovations, with the potential to complement existing glaucoma management. Ultimately, this review serves as both a valuable reference and a guidepost, steering the way toward an integrated future where IOP sensors and ophthalmology converge to enhance patient outcomes in the field of glaucoma management, all the while acknowledging the complexities and challenges that lie ahead.

## Figures and Tables

**Figure 1 micromachines-14-01915-f001:**
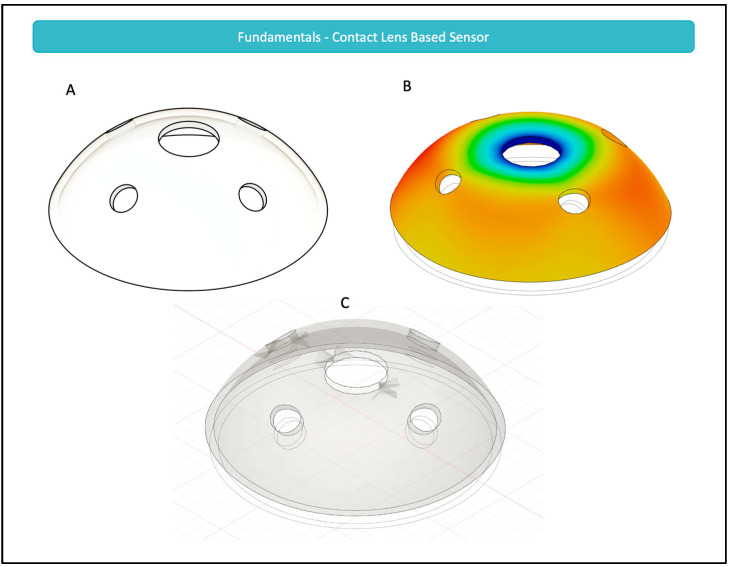
The contact-lens-based sensor: (**A**) a rudimentary design of a CLBS consisting of a central hole and four peripheral holes. The design was created on Fusion 360; (**B**) when a pressure is applied simulating an increase in intraocular pressure (IOP), the contact lens deforms, reflecting changes in the corneal curvature; (**C**) two contact lenses, the undeformed and deformed contact lens, are overlayed to show the geometrical differences in the peripheral holes. These differences in dimensions and shape can then be theoretically correlated to fluctuations in IOP.

**Figure 2 micromachines-14-01915-f002:**
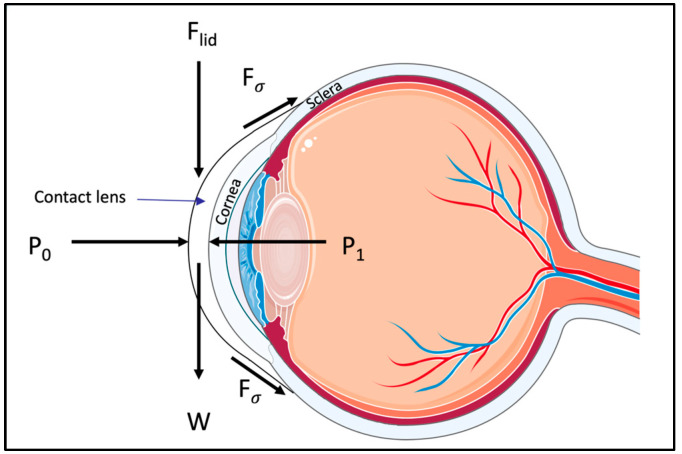
The forces acting on a contact lens. *P*_1_ is the hydrostatic pressure, which acts in opposite direction to the atmospheric pressure, *P*_0_. *F_σ_* represents the surface tension cohesive forces, *F_lid_* is the force acting on the lens when blinking, and W is the weight of the contact lens exerted due to the force of gravity. This figure was partly generated using Servier Medical Art, provided by Servier, licensed under a Creative Commons Attribution 3.0 Unported License.

**Figure 3 micromachines-14-01915-f003:**
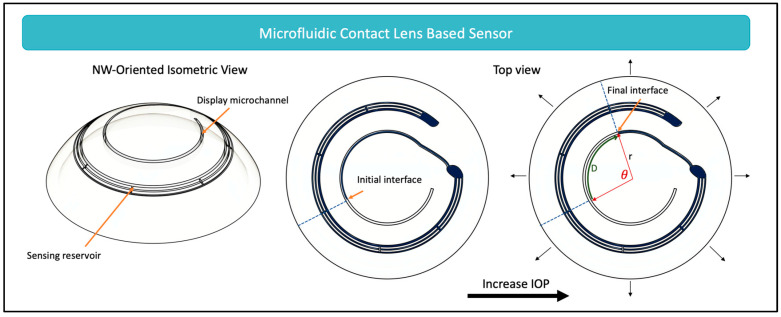
The mechanism of action of microfluidic CLBS. The sensor consists of a sensing reservoir and a display microchannel. As IOP increases, the curvature of the eye changes, resulting in an expansion of the contact lens. When the contact lens is stretched, the sensing reservoir volume increases, creating a suction effect that draws the indicator liquid–air interface towards the reservoir. Similarly, when the IOP decreases, the interface moves in the other direction—away from the sensing reservoir. Reference markings are often designed as part of the microchannel to ease the readability of the displacement and thus, enhance usability. The displacement, D is calculated as D=rθ.

**Figure 4 micromachines-14-01915-f004:**
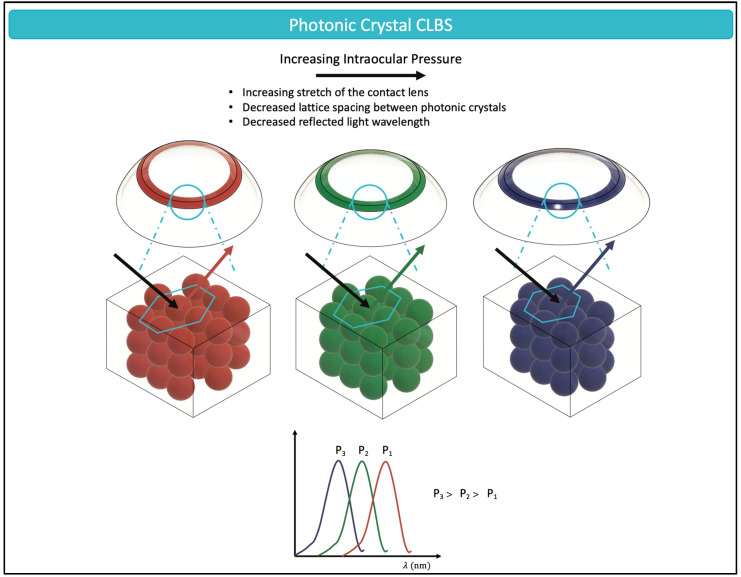
Working principle of photonic crystal CLBS. As the IOP increases (*P*_3_ > *P*_2_ > *P*_1_) and the contact lens stretches, the lattice spacing, as demonstrated by the hexagons in the figure, decreases. Consequently, and according to Bragg’s law, the wavelength of reflected light decreases resulting in a color change towards the blue hue.

**Figure 5 micromachines-14-01915-f005:**
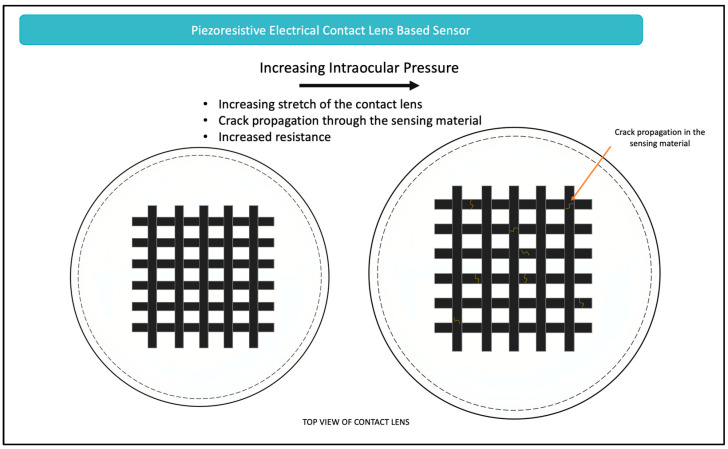
Working mechanism of piezoresistive electrical CLBS. When the IOP increases, the contact lens stretches resulting in crack propagation through the sensing material. This correspondingly results in an increase in resistance of the sensing material. This change in resistance is then correlated to IOP change.

**Figure 6 micromachines-14-01915-f006:**
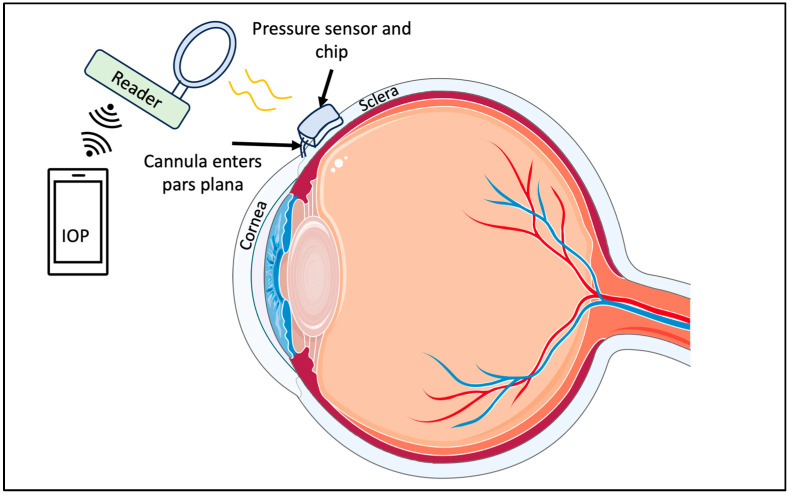
Capacitor-based implantable sensor. This sensor is designed for implantation in the superotemporal quadrant of the eye between the sclera and conjunctiva. The main components are the pressure sensor and chip, and the handheld reader with its associated radiofrequency coil. This figure was partly generated using Servier Medical Art, provided by Servier, licensed under a Creative Commons Attribution 3.0 Unported License.

**Figure 7 micromachines-14-01915-f007:**
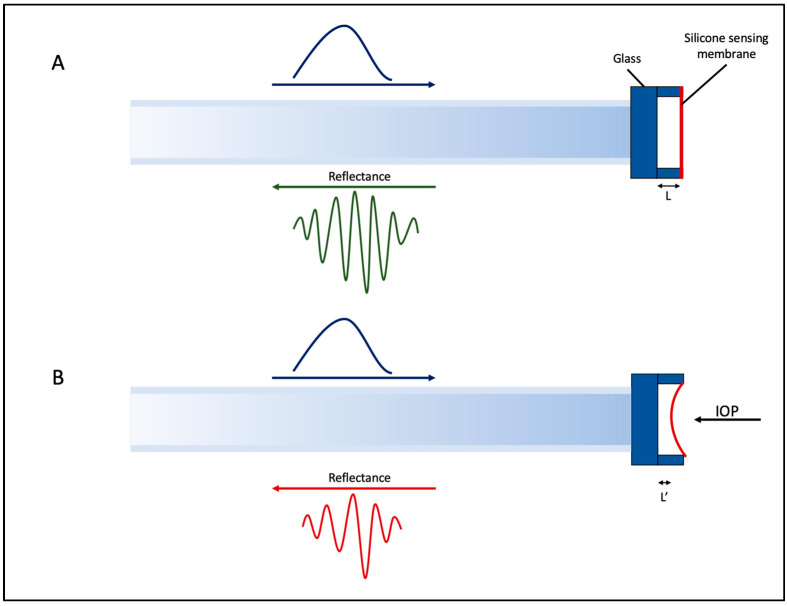
The Fabry–Perot optomechanical cavity. With increase in IOP, the fiber optic pressure sensor returns a different reflectance wavelength (**B**) as compared to undeformed cavity (**A**). This reflectance pattern is then input through an optical analyzer to determine the value of the IOP.

**Figure 8 micromachines-14-01915-f008:**
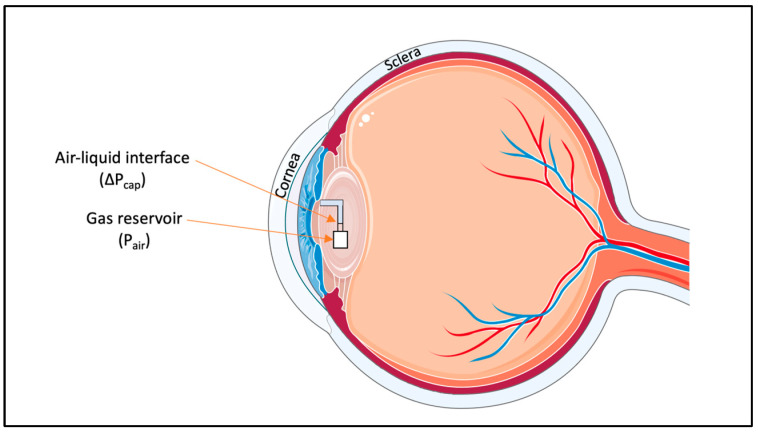
Implantable microfluidic sensor, showing a similar working mechanism to its CLBS counterpart. This figure was partly generated using Servier Medical Art, provided by Servier, licensed under a Creative Commons Attribution 3.0 Unported License.

**Figure 9 micromachines-14-01915-f009:**
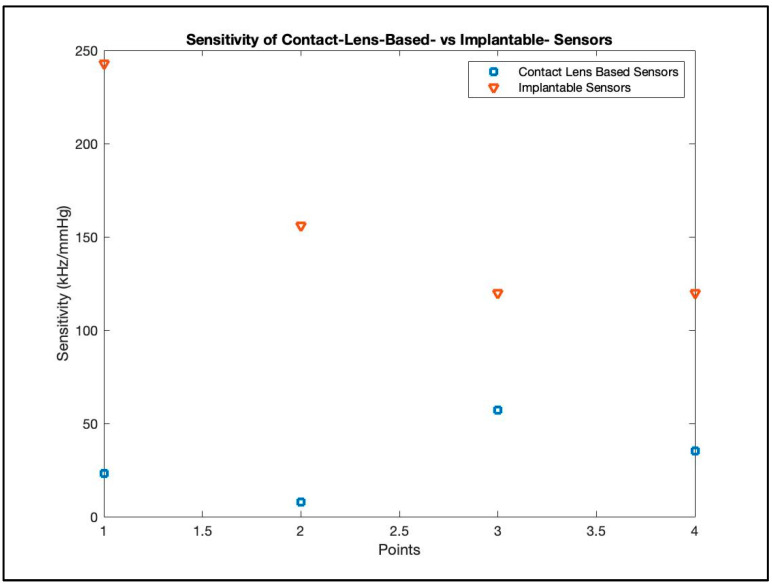
Comparison between the sensitivity of inductive couple telemetry—based on data regarding wearable IOP sensors and implantable IOP sensors obtained from [108]. The graph shows that overall, implantable sensors show better sensitivities as compared to CLBS. This understanding explains the findings in the next sections, which show that there has been commercialization of an equal number of CLBS and implantable devices. As seen in the earlier sections, when the CLBS sensitivity equals or surpasses that of implantable sensors, then typically, the implantable sensor does not flourish.

**Figure 10 micromachines-14-01915-f010:**
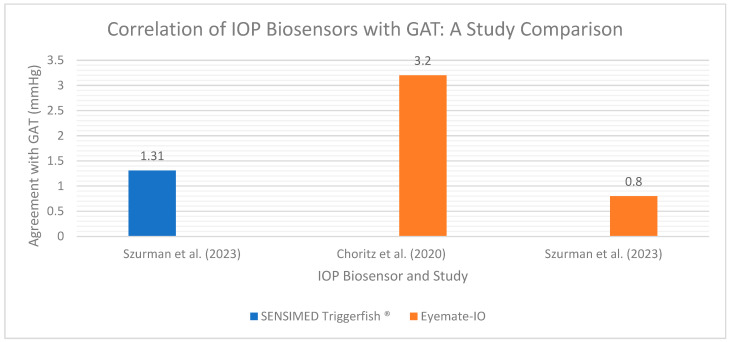
Agreement between IOP biosensors and GAT measurements across both wearable (SENSIMED Triggerfish^®^) and implantable (Eyemate-IO) sensors [124,125,133].

**Table 1 micromachines-14-01915-t001:** Comparative Analysis of Different Types of Tonometers.

Tonometer Type	Working Principle	Contact/Noncontact	Advantages	Disadvantages	Accuracy	Cost	Indication
Goldmann applanation tonometer (GAT)	Applanation	Contact	Simple Accurate Considered the gold standard	Influenced by corneal properties Requires experience Needs slit-lamp, topical anesthesia, and fluorescein	High	Low	Gold standard for IOP measurement
Perkins tonometer	Applanation	Contact	Usable in a supine position	Influenced by corneal properties Requires topical anesthesia and fluorescein	High	Low	Suitable for patients who can’t sit at the slit-lamp
Mackay-Marg–type tonometers (e.g., Tono-Pen)	Applanation	Contact	Portable Useful for patients with corneal scars or edema Usable in a supine position Does not require slit-lamp or electricity	Possible inaccuracies at high and low IOP levels Requires topical anesthesia	Moderate	Low	Patients with corneal scars or edema, supine patients
Rebound tonometers (e.g., iCare tonometer)	Ballistic probe (rebound)	Contact	Portable Easy to use Suitable for pediatrics and home tonometry No need for slit-lamp, anesthesia, or fluorescein Can be used by non-medical staff	Needs proper tip positioning Significantly influenced by central corneal thickness	High	Low	Pediatrics, home tonometry
Noncontact tonometers (e.g., air-puff, ORA)	Applanation	Noncontact	Measures the force of air to flatten the cornea ORA improves accuracy with correction algorithms Non-contact Can be used by paramedical staff	Often overestimates IOP Requires regular calibration Possible aerosol germs	Moderate (Air-Puff), High (ORA)	Medium	Large-scale glaucoma-screening programs
Dynamic contour tonometer (DCT, PASCAL)	Contour matching	Contact	Measures IOP by aligning cornea’s surface with instrument tip Believed to be less influenced by corneal properties and thickness High precision	Requires slit lamp and topical anesthesia Difficult to use Requires cooperative patients	High	Medium	After corneal refractive surgery
Pneumotonometer	Applanation	Contact	Useful for eyes with corneal scars, edema, or keratoprostheses Able to monitor IOP continuously	Influenced by corneal thickness Overestimates IOP values	Moderate	Expensive	Eyes with corneal scars, edema, or keratoprostheses
Schioetz tonometer	Indentation	Contact	Does not need electricity or slit-lamp Simple Affordable	High variability Affected by various errors Requires topical anesthesia	Moderate	Low	Primarily used in developing countries
Tactile tension	Manual pressure	Contact	Useful for uncooperative patients or detecting large IOP differences between eyes	May be inaccurate	Low	None (manual)	Uncooperative patients or for detecting large IOP differences between eyes

## Data Availability

Not applicable.

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
