# Peer review of "Advancements in Wearable and Implantable Intraocular Pressure Biosensors for Ophthalmology: A Comprehensive Review"

_micromachines, 2023, doi:10.3390/mi14101915_

Round 1

Reviewer 1 Report

This is a nicely written, well-organized, and thorough review paper on intraocular pressure sensors.  More than 200 references have been skimmed and cited.

I suggest the authors consider adding a sub-section on commercialization efforts on such sensors. While 100s of studies have been published only a handful of companies have been established.

Among them, some that come to my mind are;
- smartlens (already mentioned in the manuscript
- SCD NGoggle
- iSTAR medical (treatment implant, not sensing implant), though this might be worth mentioning.
- Glakolens Inc.
- University of Strathclyde - IDCP partnership (I'm not sure if a company has yet been established, it is worth checking)

Also, three recent relevant publications on this topic could enrich the references:

C. Johnson et al, " Comparison of Visual Field Test Measurements With a Novel Approach on a Wearable Headset to Standard Automated Perimetry" J Glaucoma, 2023

S Bhanvadia et al, "Availability of physical activity tracking data from wearable devices for patients with glaucoma", Investigative Ophthalmology & Visual Science, 2023

P. Zolfaghari et al, "MEMS Sensor–Glasses Pair for Real-Time Monitoring of Intraocular Pressure", IEEE Photonics Tech. Letters, 2023

NA

Author Response

Thank you for your suggestions and recommendations, we truly appreciate your time and review. Section 7 of our paper focuses on devices that are undergoing clinical trials and those that are in commercialization stages. While we had a few devices in that section already, we included a new subsection on Glakolens, and described the working mechanism of the contact lens-based sensor, as recommended. The other devices that are described in these sections are the Sensimed Triggerfish, the EYEMATE by Implandata, and the IOP connect device by Injectsense. All of these devices are either in testing or commercial stages. Additionally, the Smarlens Inc device is described in section 5.1.1.

The other devices that you have recommended for us to include, we have researched extensively. The SCD NGoggle is a Virtual Reality headset that does perimetry testing to monitor the progression of glaucoma by measuring the visual field defects over time. The device was found to be outside the scope of the paper, since it does not relate to the measurement of intraocular pressure. Rather, the device measures visual field defects; there are review papers that specifically cover this topic due to its grandiosity.

Similarly, iSTAR Medical, which as described is a treatment implant, is also determined to be outside the scope of this review article. There are many treatment devices - if one is included, many others would also meet inclusion criteria, which would consequently take away from the objective of the paper.

Finally, no evidence of the commercialization of the University of Strathclyde - IDCP partnership was found.

Two of the three references provided were included, while the one discussing the Virtual Reality headset (by C. Johnson et al.) was not included for the same reasons as discussed previously.

Reviewer 2 Report

The authors presented a thorough review IOP sensors. Besides the IOP sensors, the authors included in-depth review on glaucoma and clinical treatment, etc, which is fairly beneficial to readers. 

The authors covered some recently developed CLS.  The paper can be improved in the IOP sensor review part, if the authors can include more CLS or implanted sensors reported. BJO, EYE,  JAMA, ARVO, etc. had many related papers.

Author Response

Thank you for your review and recommendations. We have included a subsection on Glakolens, a novel CLS and described its mechanism of action. Additionally, we included a description of an interferometric implantable micro-electromechanical systems (MEMS) sensor (lines 526-531).

All in all, we have discussed the majority of the IOP sensors in the literature that have been described in the past five years. Older devices may have been excluded to ensure that our review is current and comprehensive. However, even in those cases, we made sure to describe the theory and design basics for each different kind of sensor.

Thank you again for your comprehensive review.

Round 2

Reviewer 1 Report

The authors have addressed my comments.